TECHNICAL RELEASE

# Nanopore adaptive sampling enriches for antimicrobial resistance genes in microbial communities

Danielle C. Wrenn[1] and Devin M. Drown[1,2,*]

1 Department of Biology and Wildlife, University of Alaska Fairbanks, Fairbanks, Alaska, USA
2 Institute of Arctic Biology, University of Alaska Fairbanks, Fairbanks, Alaska, USA

## ABSTRACT

Antimicrobial resistance (AMR) is a global public health threat. Environmental microbial communities act as reservoirs for AMR, containing genes associated with resistance, their precursors, and the selective pressures promoting their persistence. Genomic surveillance could provide insights into how these reservoirs change and impact public health. Enriching for AMR genomic signatures in complex microbial communities would strengthen surveillance efforts and reduce time-to-answer. Here, we tested the ability of nanopore sequencing and adaptive sampling to enrich for AMR genes in a mock community of environmental origin. Our setup implemented the MinION mk1B, an NVIDIA Jetson Xavier GPU, and Flongle flow cells. Using adaptive sampling, we observed consistent enrichment by composition. On average, adaptive sampling resulted in a target composition 4× higher than without adaptive sampling. Despite a decrease in total sequencing output, adaptive sampling increased target yield in most replicates. We also demonstrate enrichment in a diverse community using an environmental sample. This method enables rapid and flexible genomic surveillance.

**Subjects** Software and Workflows, Bioinformatics, Metagenomics

## STATEMENT OF NEED

Antimicrobial resistance (AMR) is a public health threat of great magnitude, accounting for over 2.8 million infections and 35,000 deaths annually in the USA alone [1]. Resistant pathogens pose the most direct risk to human health. However, the AMR genes present in pathogens only represent a small proportion of a much larger collection of AMR genes: the antibiotic resistome. As described by Wright [2] and D'Costa et al. [3], the antibiotic resistome includes all antimicrobial resistance genes and their precursors, the majority of which reside in nonpathogenic microbial communities.

Environmental microbial communities are important contributors to the resistome. They are dynamic reservoirs where a variety of factors influence the evolution, exchange, and persistence of genes that confer resistance. Resistance mechanisms originated in the environment [2, 4], where the production of and the resistance to antimicrobial agents assist microorganisms in their battle for territory and resources [4, 5]. External factors, like human and agricultural waste streams, introduce resistant organisms, resistance genes, and pharmaceutical antimicrobial agents into the environment [6, 7]. Once within the community, these agents provide additional selective pressure for resistance, and the genes provide new material for exchange.

**Submitted:** 03 July 2023

* Corresponding author. E-mail: dmdrown@alaska.edu

Preprint submitted at https://doi.org/10.1101/2023.06.27.546783

The exchange of resistance genes between community members continues even without selective pressure [8]. This continued exchange is likely one reason why antimicrobial resistance can persist even after the removal or reduction of antimicrobial exposure [9, 10]. The likelihood of a resistant community reverting to a susceptible one is a complex landscape influenced by mutation rate, fitness cost, and compensatory evolution [11].

AMR genes can then be shared from environmental microbial communities to human and animal pathogens [12, 13]. The One Health approach, which recognizes the interconnection of human, animal, and environmental health, has grown in popularity regarding microbiology and AMR research [14–17]. The World Health Organization [18] and the Center for Disease Control and Prevention [1] have both endorsed the One Health approach as an effective strategy for addressing AMR.

Despite the increased interest in investigating environmental AMR, gaps in knowledge still exist regarding the exchange of AMR genes between environmental organisms and pathogenic communities, the effects of abiotic factors on the persistence and evolution of environmental AMR, and the effects of clinical and agricultural interventions on environmental microbial communities. Genomic surveillance of genes associated with AMR could provide important insight into how these dynamic reservoirs impact public health. Genomic surveillance allows for monitoring the entire resistome, encompassing AMR genes inside and outside pathogenic organisms, as well as their precursors. It also allows for detecting "silent" AMR genes – those present in susceptible organisms but potentially conferring resistance following a shift in host or environment. However, genomic sequencing is typically time, resource, and cost-intensive, especially outside clinical settings.

Nanopore sequencing presents an opportunity to develop a cost-effective and portable genomic surveillance tool. While more commonly used sequencing technologies sequence via DNA synthesis, nanopore sequencing determines genetic sequences by detecting a change in current as DNA strands are pulled through nanopores on the flow cell [19]. The technology allows for a streamlined, resource-conservative library preparation. It also allows for unique features like adaptive sampling [20, 21].

Traditional sequencing technologies, such as Illumina, achieve enrichment by using reactions such as PCR prior to sequencing. Pre-sequencing enrichment necessitates additional time and resources, including synthesized primers. In contrast, adaptive sampling requires no change in library preparation as it leverages the ability of each nanopore to independently accept and reject strands of DNA during sequencing. The enrichment or depletion of user-defined targets is therefore achieved entirely *in silico*, without the need for additional time, resources, or effort. The MinION, the smallest genomic sequencer currently commercially available, boasts incredible portability (with minimal power consumption) in addition to being capable of adaptive sampling.

Other studies used nanopore sequencing and the adaptive sampling feature to detect AMR genes in clinical samples through both host depletion [22] and AMR gene enrichment [23]. However, the exploration of adaptive sampling for the enrichment of AMR genes in environmental metagenomic samples is limited.

Here, we developed a novel toolbox optimized for the rapid, resource-conservative surveillance of AMR-associated genes in environmental microbial communities. The principal question addressed by our study was whether adaptive sampling can enrich (by composition) for AMR-associated genes in a mock community of environmental origin. This study investigated performance metrics, including enrichment by target yield and the proportion of the panel that was successfully detected.



## IMPLEMENTATION

### Methods

#### Experimental design

To test the effects of adaptive sampling on AMR gene enrichment, we included two treatments: adaptive sampling 'on' and 'off'. We simultaneously implemented these two treatments by turning on adaptive sampling for 50% of the sequencing nanopores on the flow cell while the other half sequenced the library using the traditional, non-selective method (adaptive sampling 'off'). With this design, we could control for variability in our library preparation from run to run.

We generated a mock community from bacterial isolates with known AMR genes from previously isolated and archived soil samples from the Fairbanks Permafrost Experiment Station [24]. The original bacterial culturing and isolation methods are described by Haan and Drown *et al.* [24]. To compose our final mock community, we selected six community members (TH25, TH28, TH41, TH57, TH79, and TH81) representing five genera (*Serratia*, *Bacillus, Erwinia, Pantoea,* and *Pseudomonas*) of common soil bacteria associated with permafrost thaw. These members were selected to achieve a phylogenetic diversity, including a diverse set of AMR genes. For this experiment, we extracted and purified DNA from previously frozen cells using the DNeasy UltraClean Microbial Kit (Qiagen) according to the manufacturer's instructions. After quantifying the DNA concentration from the extractions using a Qubit (Thermo Fisher Scientific), we pooled all members of the community by equal mass (1000 ng).

Using published sequences (Biosample accessions SAMN17054805, SAMN17054834, SAMN17054856, SAMN09840060, SAMN17054818, and SAMN17054803) [24], we identified all AMR gene regions using the Resistance Gene Identifier (RGI) version 5.1.0 and the Comprehensive Antibiotic Resistance Database (or CARD) [25] version 3.0.9. The target gene panel was constructed using exclusively strict and perfect hits. Targeted genes and the number of gene copies per community member are specified in Table 1. The expansion of targeted regions through the inclusion of flanking DNA was implemented by previous studies [20, 26] and is recommended by Oxford Nanopore to increase the target output. We used a custom script to expand the gene region and include a flanking region of DNA in each targeted region. See Figure 1 for an overview of the bioinformatic worfklow, and GigaDB for the custom scripts [27]. Each flanking region was the size of the prepared library's N50 (5,075 bp). Due to the fragmentation of the available genome assemblies, not all target regions could be expanded to the entire length of the flanking region on both sides. Each target region was expanded as far as possible to a maximum of 5,075 bp of additional genomic material on either side. We extracted target sequences using Geneious Prime 2022.1.1 (RRID:SCR_010519) [28]. The resulting multi-fasta file contained 52 unique sequences and served as the adaptive sampling reference.

#### Library preparation and sequencing

We used the Rapid Sequencing Kit (SQK-RAD004) of Oxford Nanopore Technologies to prepare the sequencing libraries. For each library, we used 200 ng of input DNA from our mock community. We followed the manufacturer protocol, except we excluded the bead cleanup and the Qubit quantification steps to maximize the DNA quantity carried forward into sequencing.

**Table 1.** Targeted genes.

| Member | TH25 | TH28 | TH41 | TH57 | TH79 | TH81 |
|---|---|---|---|---|---|---|
| Genus | *Bacillus* | *Serratia* | *Pseudomonas* | *Bacillus* | *Erwinia* | *Pantoea* |
| AMR Gene | | | | | | |
| BcII | 1 | 0 | 0 | 1 | 0 | 0 |
| FosB | 1 | 0 | 0 | 1 | 0 | 0 |
| MCR-4.5 | 1 | 0 | 0 | 0 | 0 | 0 |
| tet(45) | 1 | 0 | 0 | 0 | 0 | 0 |
| CRP | 0 | 1 | 0 | 0 | 1 | 1 |
| *Escherichia coli* EF-Tu mutants conferring resistance to Pulvomycin | 0 | 1 | 0 | 0 | 0 | 0 |
| *Haemophilus influenzae* PBP3 conferring resistance to beta-lactam antibiotics | 0 | 1 | 0 | 0 | 1 | 1 |
| *Klebsiella pneumoniae* KpnF | 0 | 1 | 0 | 0 | 1 | 1 |
| *Klebsiella pneumoniae* KpnH | 0 | 1 | 0 | 0 | 1 | 1 |
| adeF | 0 | 5 | 3 | 0 | 2 | 2 |
| emrR | 0 | 1 | 0 | 0 | 1 | 1 |
| msbA | 0 | 1 | 0 | 0 | 1 | 1 |
| *Acinetobacter baumannii* AbaQ | 0 | 0 | 1 | 0 | 0 | 0 |
| *Pseudomonas aeruginosa* soxR | 0 | 0 | 1 | 0 | 0 | 0 |
| armA | 0 | 0 | 1 | 0 | 0 | 0 |
| MCR-4.1 | 0 | 0 | 0 | 1 | 0 | 0 |
| sgm | 0 | 0 | 0 | 1 | 0 | 0 |
| CARB-23 | 0 | 0 | 0 | 0 | 1 | 0 |
| *Escherichia coli* ampH beta-lactamase | 0 | 0 | 0 | 0 | 1 | 1 |
| *Morganella morganii* gyrB conferring resistance to fluoroquinolone | 0 | 0 | 0 | 0 | 1 | 1 |
| PmrF | 0 | 0 | 0 | 0 | 1 | 0 |
| BES-1 | 0 | 0 | 0 | 0 | 0 | 1 |
| *Escherichia coli* UhpT with mutation conferring resistance to fosfomycin | 0 | 0 | 0 | 0 | 0 | 1 |
| *Klebsiella pneumoniae* KpnE | 0 | 0 | 0 | 0 | 0 | 1 |
| amrB | 0 | 0 | 0 | 0 | 0 | 1 |

The MinION mk1B, an NVIDIA Jetson Xavier GPU, and flongle flow cells (FLO-FLG001, R9.4.1) were used for sequencing. We configured the Xavier GPU with MinKNOW (MinKNOW Core version 4.5.4) following the instructions from Benton [29]. All sequencing runs lasted eight hours. Using MinKNOW, we designated half of the flow cell (63 channels) for adaptive sampling; the other half of the flow cell sequenced normally (adaptive sampling 'off'). This setting is in the Run options under 'Advanced options'. We alternated the side of the flow cell, performing each treatment for each replicate. Each flow cell was used twice (technical replicates) by starting a new sequencing run after eight hours without washing the flow cell and using the same initial library. We completed thirteen total sequencing runs.

### Data and statistical analysis and visualization

We used Guppy version 6.1.3 (RRID:SCR_023196) to base call the raw sequencing data using the super-accuracy model (dna_r9.4.1_450bps_sup.cfg) and filtered by minimum quality score (Q score ≥ 10). During the adaptive sampling, the first 500–1000 bp of a template strand of DNA were sequenced. Regardless of the decision of the adaptive sampling algorithm (accept or reject), that preliminary sequence was the output. To remove these very short reads, we filtered the output by length (>1000 bp) using Seqtk version 1.3 (RRID:SCR_018927) [30]. We aligned the filtered output to the community metagenome with

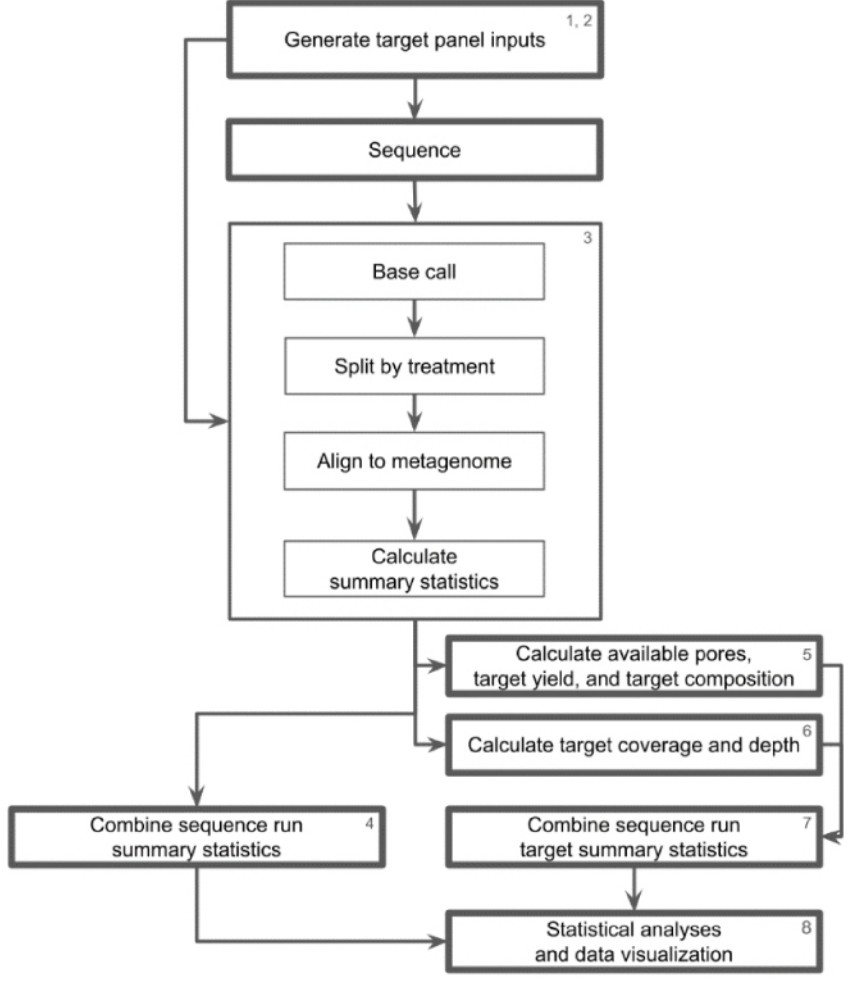

**Figure 1.** Overview of bioinformatic workflow. At each step, we used the following scripts available in GigaDB: (1) generating_target_panel_files.R, (2) generating_environmental_target_panel_files.R, (3) dart_methods_notebook. md, (4) generating_multi_run_nanostats_csv.R, (5) generating_single_run_analysis_files.R, (6) generating_single_ run_depth_csv.R, (7) generating_multi_run_analysis_files.R, (8) statistical_analysis_data_viz.R. See also the dart_methods_notebook.md file bringing all of the scripts and their parameters together [27].

Minimap2 version 2.22 (RRID:SCR_018550) [31] using the Oxford Nanopore genomic reads preset (-ax map-ont). We used SAMtools version 1.15.1 (RRID:SCR_002105) to exclude supplementary and secondary alignments (-F 2308) [32]. We used the sequencing summary generated by Guppy to calculate the average number of active pores. We first subset the data by treatment, then binned the data into one-hour intervals. The number of unique channels generating reads was then calculated for each hour and averaged across the run length.

We calculated the summary statistics for each run (yield and mean quality score) using NanoStat version 1.6.0 [33]. To calculate the target yield, we used SAMtools coverage and depth. SAMtools coverage was implemented to determine the number of reads that contained targeted AMR regions; depth was used to determine the number of nucleotides that aligned to targeted AMR regions. In these calculations, AMR regions referred to the AMR genes without expanded flanking regions. To avoid a single read being counted

multiple times in our target yield calculations, we only included unique alignments in downstream analysis.

We utilized base R (version 4.2.2; RRID:SCR_001905) and the R car package [34] for statistical analyses. We used the Shapiro-Wilk test to determine data normality. Variance homogeneity was determined using either an *F*-test or Levene's test, as appropriate. A Two-Sample *t*-test, Welch's *t*-test, or Wilcoxon signed-rank test was then employed to evaluate the significance of any difference between treatments ($\alpha$ = 0.05). For data visualization, we used ggplot2 (RRID:SCR_014601) [35].

### Environmental sample

In order to demonstrate the potential effectiveness of these methods on a more diverse community, we applied our methods to a microbial community from soil. We first sequenced the community without using adaptive sampling to identify AMR genes potentially present in the microbial community. We then used our previously described methods to test the ability to enrich for these AMR targets using adaptive sampling.

The soil microbial community came from a 10 m transect in remote Alaska (66.792436° N, 160.49554° W). Ten cores with a 2.9 cm diameter were collected using a sterile technique and a soil probe to obtain the top 10 cm of soil. We extracted total genomic DNA from 250 mg of soil per homogenized soil core using the DNeasy PowerSoil Pro kit (Qiagen; Germany) following manufacturer instructions. We used the Native Barcoding Kit (SQK-NBD114.24) for sequencing library preparation to multiplex ten samples. We sequenced the library using a MinION (MinKNOW Core version 5.4.7) on an R10.4.1 Flow cell (FLO-MIN114) for 72 hours.

Following sequencing, we base called the raw sequencing data with Guppy version 6.5.7 using the super-accuracy model (dna_r10.4.1_e8.2_400bps_sup.cfg) and filtered by minimum quality score (Q score ≥ 10). We initially used the RGI version 6.0.2 to classify reads with AMR open reading frames. We used BLAST (RRID:SCR_004870) for alignment (-a BLAST) and the –low_quality and –include_nudge options to include partial AMR genes and low-quality matches. Using the output from RGI, we curated our high-quality target panel by excluding nudged matches and including only strict and perfect hits. A custom script was then used to expand the target region to include flanking DNA. Each flanking region was the N50 (3180 bp) of the prepared library. Due to the lack of complete genome assemblies for community members, some target regions could not be expanded to the full flanking region length. As a result, each target region was expanded as much as possible to a maximum of 3,180 bp of additional genomic material on either side of the target. The target regions were then extracted using Seqtk (version 1.3-r106).

For the adaptive sequencing run of this environmental sample, we created a pool of DNA from the ten soil cores. We prepared a library using the Rapid Sequencing Kit (SQK-RAD004) and sequenced the library using the same parameters as the mock community experiments, with the modification of using an R9.4.1 flow cell (FLO-MIN106D).

### Results

Over the course of four days, we completed 13 sequencing runs of the mock community. We excluded three runs lacking pores at the end of the first technical replicate. This resulted in ten sequencing runs, including technical replicates, that were used for our analysis (Table 2). Our maximum output run generated over 281 Mb of data, the lowest over 19 Mb.

**Table 2.** Sequencing output metrics prior to and following filtering for quality and read length.

| Run | Technical replicate | Pre-Filtering | | Post-Filtering | |
|---|---|---|---|---|---|
| | | Yield (bp) | Mean quality (Q) Score | Yield (bp) | Mean quality (Q) Score |
| 1 | 1 | 281,215,264 | 10.3 | 109,639,119 | 12.3 |
| 2 | 2 | 98,850,455 | 8.2 | 18,970,458 | 11.4 |
| 3 | 1 | 170,831,418 | 10.6 | 86,179,378 | 12.5 |
| 4 | 2 | 77,078,419 | 9.4 | 22,621,178 | 12.2 |
| 5 | 1 | 99,711,784 | 10.5 | 40,575,094 | 12.8 |
| 6 | 2 | 51,020,783 | 9.1 | 20,600,281 | 12.5 |
| 7 | 1 | 146,519,076 | 11.3 | 62,208,499 | 13.3 |
| 8 | 2 | 38,318,184 | 10.8 | 21,961,623 | 12.5 |
| 9 | 1 | 54,700,216 | 11.5 | 32,243,307 | 12.8 |
| 10 | 2 | 19,037,961 | 9.8 | 6,764,271 | 12.0 |

On average, sequencing runs yielded 103,728,356 bp and contained 42,176,321 bp after filtering by quality and length. On average, second technical replicates generated 62% less data ($\mu_{first}$ = 150.6 Mbases, $\mu_{second}$ = 56.86 Mbases) and a lower mean output quality (a decrease of 12.7%). Filtering by quality and length resulted in a 59% decrease in yield but a 23% increase in quality (Table 2). Only post-filtering data were used in alignment and target yield quantification.

Regardless of sequence identity, we observed a significant decrease in sequencing output when using adaptive sampling ($t$ = −6.67, $p$ = 2.968 × 10$^{-6}$) (Figure 2). Although the adaptive sampling 'off' treatment showed greater variability in output between runs ($\sigma^2$ = 1.09) compared to when adaptive sampling was 'on' ($\sigma^2$ = 0.42), this difference was not statically significant ($F$ = 0.385, $p$ = 0.171). Here, sequencing output refers to the total sequencing yield (pre-filtering) per treatment. While we split the flow cell evenly across treatments, there might have been variation in pore availability between flow cells and treatments. To control for this variation, we normalized these yields by the average number of active pores during the sequencing run. The need for this normalization was compounded by our use of technical replicates, where we saw an increase in the variation of active pores.

Next, we evaluated AMR gene target enrichment by composition. This is a measure of the fraction of the sequencing output that includes the targeted AMR genes. To this purpose, we calculated the percent target composition for each treatment and sequencing run, where percent composition was calculated as follows: (Output aligned to target AMR genes (bp)/Total pre-filtering sequencing output (bp)) ∗ 100. Despite the decreased yield observed in the adaptive sampling treatment (Figure 3), the proportion of sequencing output composed of target AMR genes was significantly greater for the adaptive sampling treatment ($V$ = 55, $p$ = 0.002). On average, the percent target composition achieved by adaptive sampling was over 4× higher than that observed in the control treatment (Figure 4). We found that over 0.42% of the output of the adaptive sampling treatment represented the target gene sequences, on average. For context, we estimate that the true representation of the targeted AMR genes in our sample metagenome is 0.24%.

We also evaluated enrichment by target yield. This is a measure of the sequencing yield (Kbases) that was solely composed of the designated target genes. To measure the performance difference between treatments, we calculated the percent difference between treatments for each of our sequencing runs as normalized by the control run. The percent difference was calculated as follows: [(target yield (bp) per average active pores with

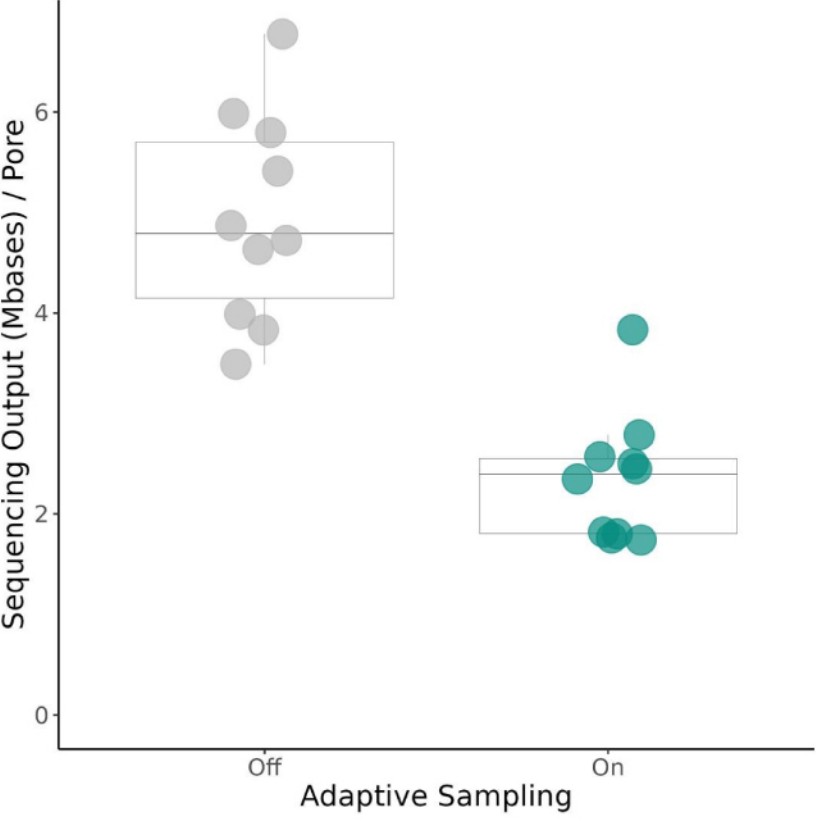

**Figure 2.** A comparison of total sequencing output with and without the use of adaptive sampling. Sequencing output refers to all the data generated prior to filtering for quality and length. Total output is normalized using the average number of active pores during the entire run duration for each treatment. Statistical analysis used a paired Welch's $T$-test ($t = -6.67$, $p = 2.968 \times 10^{-6}$). $\mu_{\text{OFF}} = 4.95$ Mb, $\mu_{\text{ON}} = 2.36$ Mb ($n = 10$).

adaptive sampling − target yield (bp) per average active pores without adaptive sampling)/target yield (bp) per average active pores without adaptive sampling] ∗ 100. Positive values indicated that using adaptive sampling resulted in a greater target yield. The difference in target yield was significantly greater than zero ($V = 54$, $p = 0.00195$) (Figure 5). Adaptive sampling outperformed the control treatment in this metric for nine out of ten replicates. The mean percent difference between the two treatments was 104.6%, representing a greater than two-fold increase in target yield when adaptive sampling was used (Figure 5).

Finally, we looked at the proportion of our target panel detected by each treatment. Our criteria for detection were as follows: 100% coverage of the AMR region with a minimum depth of 2 nucleotides at every position. Due to output requirements inherent in the criteria, sequencing runs that generated less than 25 Mb of post-filtering data were excluded from this analysis ($n = 5$). When adaptive sampling was used, 21.9% of the panel was detected, on average. This is more than double the average 8% observed when adaptive sampling was not used (Figure 5). The maximum proportion detected was 36.5% and 21.2% with adaptive sampling 'on' and 'off', respectively. Within a sequencing run, the side of the flow cell implementing adaptive sampling consistently detected more of the panel than its

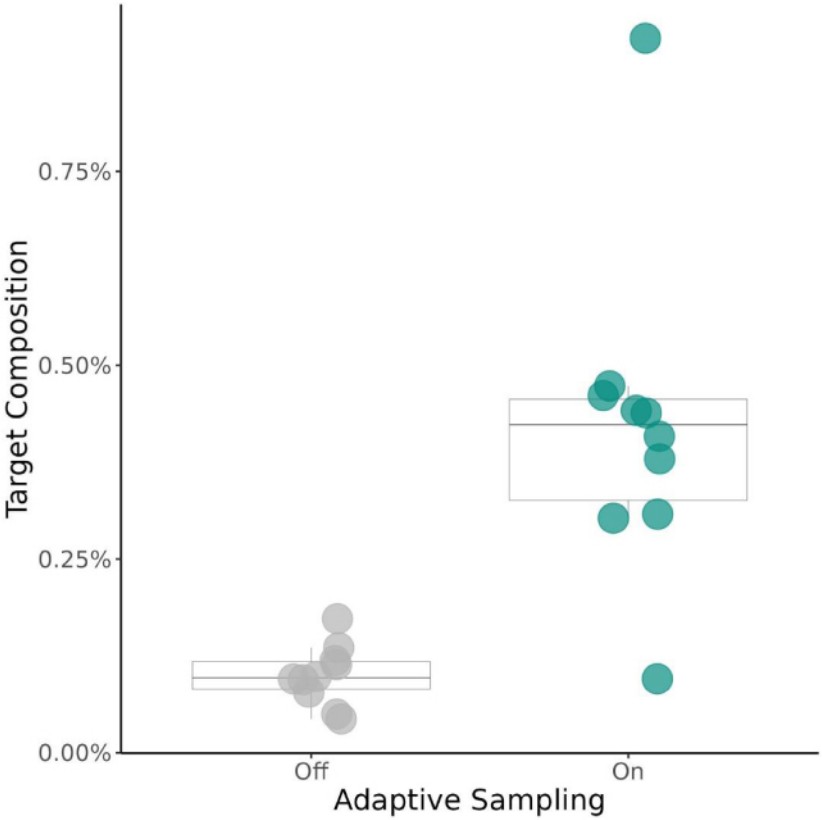

**Figure 3.** Comparison of the target composition of total sequencing output with and without the use of adaptive sampling. Percent target composition was calculated as the output aligned to a targeted AMR region (bp)/total sequencing output (pre-filtering) (bp) ∗ 100. Statistical analysis used a Wilcoxon signed-rank test ($V = 55$, $p = 0.002$). $\mu_{OFF} = 0.1\%$, $\mu_{ON} = 0.42\%$ ($n = 10$).

non-adaptive sampling counterpart; however, we did not find the difference between these two treatments to be significant ($V = 15$, $p = 0.0625$).

### Environmental sample example

We applied our method to a diverse soil microbial community. We characterized the known AMR gene targets without adaptive sampling and using a high-yield ligation-based sequencing kit. This sequencing yielded 14,041,647,517 bp. Using RGI, we identified 943 high-quality gene targets, totaling 4,757,091 bp in our target database, after including flanking sequences. Next, we sequenced the community again following our adaptive sampling methods and splitting the flow cell across the two treatments (adaptive sampling 'on' and 'off'). We generated 1,066,363,786 bp (mean Q score 11.3) before filtering and 595,473,739 bp (mean Q score 13.3) of post-filtered data. The flow cell used for the environmental sample was old, likely contributing to the lower total output. Similarly to our first sequencing runs, we observed a lower sequencing output when using adaptive sampling than without (3.08 Mbases/pore vs 6.71 Mbases/pore). However, despite the lower yield, the proportion of sequencing output composed of target AMR genes was greater for the adaptive sampling treatment. We found that over 0.026% of the output of the adaptive sampling treatment represented the target gene sequences, in contrast to the control



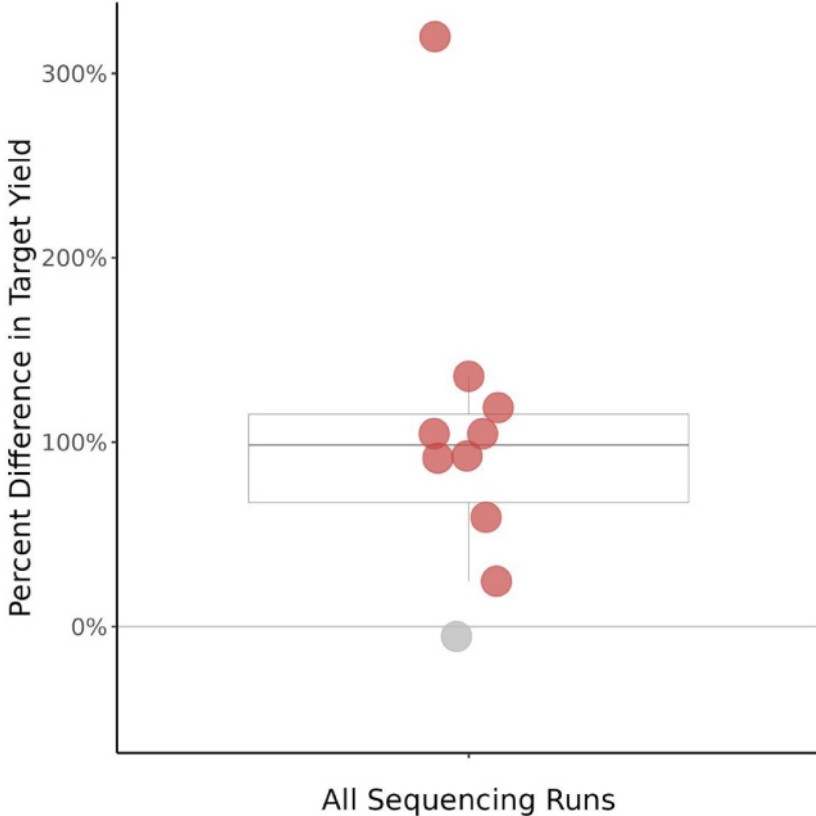

**Figure 4.** The percent difference in target yield between adaptive sampling on and adaptive sampling off sides of each flow cell. Percent difference was calculated with the half of the flow cell sequencing normally (adaptive sampling off) as the initial value. Red points denote a difference >0%, gray points denote a difference ≤0%. Statistical analysis used a Wilcoxon signed-rank test ($V = 54$, $p = 0.00195$). $\mu = 104.6\%$ ($n = 10$).

treatment's 0.011%. Additionally, we evaluated the enrichment by target yield. The percent difference between the two treatments was 11.12%, representing a greater than 1.11-fold increase in target yield when adaptive sampling was used. No target regions met our criteria for detection (2× coverage) in either treatment.

## Discussion

This research represents the first steps in developing a novel toolbox for the rapid, resource-conservative surveillance of AMR-associated genes in environmental microbial communities. Our goal in this study was to assess the ability of adaptive sampling to enrich (by composition) for AMR-associated genes in a known sample. We found that adaptive sampling could enrich for AMR genes in our mock community. We observed consistent enrichment by composition when using adaptive sampling regardless of the overall sequencing yield. When applied to a diverse microbial community from an environmental source, adaptive sampling also enriched for antimicrobial resistance genes. While Martin *et al.* [36] demonstrated the ability of adaptive sampling to enrich by composition for genomes in metagenomic samples, here we demonstrated that adaptive sampling can enrich for much smaller targets – i.e., AMR genes in microbial communities. Our observations regarding enrichment by target yield are encouraging. Other studies have

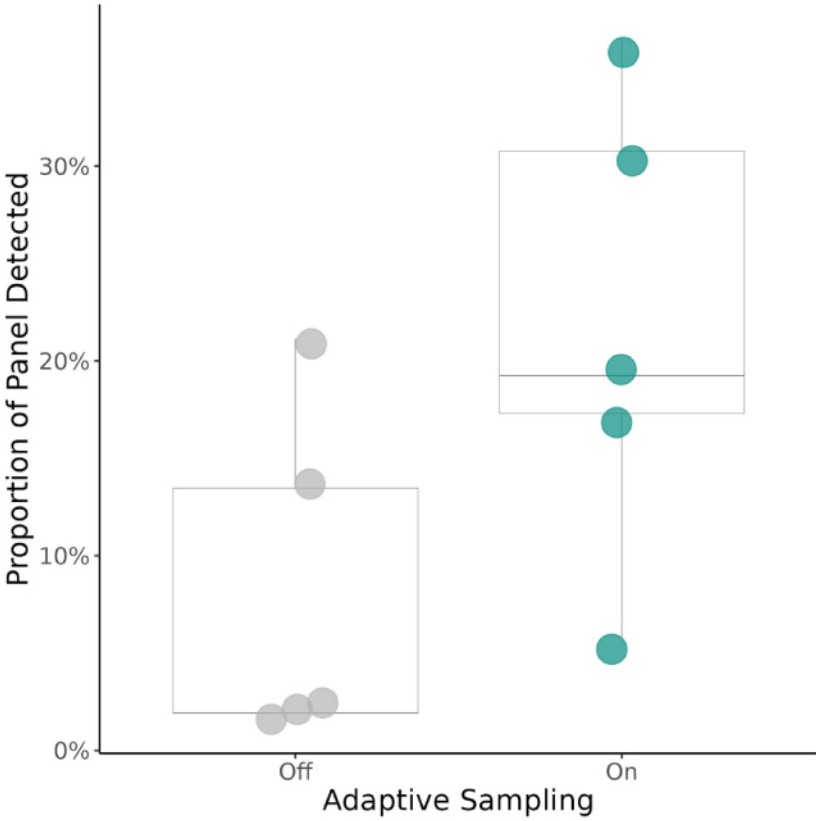

**Figure 5.** Proportion of the target AMR gene panel detected for adaptive sampling on and off treatments. A successful detection was defined as 100% AMR gene coverage with ≥2 bp depth at every position. Statistical analysis used the Wilcoxon signed-rank test ($V = 15$, $p = 0.0625$). $\mu_{OFF} = 8.1\%$, $\mu_{ON} = 21.9\%$ ($n = 5$).

noted the association between enrichment by yield and sequencing run output [36]. This is due, in part, to the variability in pore quality and pore loss between flow cells. Our use of technical replicates, where second technical replicates began with fewer available pores and those that remained were likely decreased in quality, may have further exacerbated this effect in our study.

Further optimization could increase enrichment by yield using adaptive sampling. The available literature suggests that template length, target size, percent identity, and the above-mentioned pore availability can all impact enrichment by yield [23, 36]. The ratio of target size to template length affects the likelihood of the pore detecting the target sequence before the algorithm rejects that strand. Small targets on long templates have a higher likelihood of being missed. The lower the percent identity between the target and template also increases the likelihood that a sequence will not be recognized as on-target [23]. This is due to adaptive sampling's reliance on the live alignment of template strands to target sequence data to determine target presence. Finally, pore quality and availability directly impact the sequencer's ability to generate both on-target and off-target data [36].

For these experiments, we relied on low-cost Flongle flow cells that cost a fraction of the cost of a traditional flow cell while generating a fraction of the yield. However, the combination of low target yield and overall low sequencing output contributed to the

inability of either treatment to detect more than 37% of our target panel. Consequently, optimization in target yield may improve panel detection. We employed increased target size in the pursuit of greater enrichment by yield. Further work is needed to explore the employment of other strategies to produce consistent enrichment by target yield in our protocol.

The expansion of current knowledge regarding resistance in environmental microbial communities benefits the One Health approach to addressing the threat of AMR. Environmental microbial communities play an important role in the origin, persistence, and dissemination of resistance mechanisms [2, 4, 9, 10]. Unlike our mock community, environmental communities tend to be highly diverse with an uneven abundance of community members. Even with this challenge, our environmental sample example provided modest evidence that enrichment for small targets in a diverse microbial community can be achieved. The MinION, with its incredible portability and ability to perform adaptive sampling, could reduce time-to-answer and economic barriers to genomic surveillance of environmental reservoirs of AMR-associated genes.

Other studies have described the potential benefit of using adaptive sampling to reduce time-to-diagnosis in clinical samples [22]. Reduced time-to-answer in an environmental context could allow for better informed preventative public health action, industry-standard modification, and policy implementation. The scope of this study was limited in terms of communities tested and AMR genes targeted. However, its results are promising for developing a flexible, portable, and cost-effective AMR surveillance tool. Future work could include expanding the target gene panel to allow the toolbox to be applied to a larger cohort of microbial communities and conducting thorough testing of the protocol on diverse environmental samples.

## DATA AVAILABILITY

The datasets supporting the results of this article are available in the SRA database under BioProject PRJNA982864 and PRJNA1031672. Other data and R-scripts are available in GigaDB [27].

## LIST OF ABBREVIATIONS

AMR: antimicrobial resistance; RGI: Resistance Gene Identifier.

## DECLARATIONS

### Ethical approval

The authors declare that ethical approval was not required for this type of research.

### Competing interests

DCW has received funding for travel, accommodation, and conference fees to speak at events organized by Oxford Nanopore Technologies.

### Authors' contributions

DCW, DMD: conceptualization, investigation, formal analysis, software, methodology, validation, data curation, resources, funding acquisition, visualization. DCW: original draft preparation. DCW, DMD: review and editing.

## Funding

This work was supported by Alaska BLaST and Alaska INBRE. BLaST is supported by the NIH Common Fund, through the Office of Strategic Coordination, Office of the NIH Director with the linked awards: TL4GM118992, RL5GM118990, and UL1GM118991. Alaska INBRE is supported by an Institutional Development Award (IDeA) from the National Institute of General Medical Sciences of the National Institutes of Health under grant number P20GM103395.

## Acknowledgements

We would like to thank Tracie Haan for supplying the isolates used in our mock community and for technical support. We would also like to thank Bevyn Cover for providing the DNA and the initial sequence data for the environmental sample. We would like to thank Upasana Arora, Jeremy Buttler, Bevyn Cover, Ursel Schütte, and Jorda Kovash for their constructive feedback on the project design. We would like to thank Miles Benton, who kindly provided detailed instructions for computer setup and inspiration for this project. We thank the reviewers for their constructive feedback on the manuscript. We acknowledge the generous support of the Institute of Arctic Biology and Logan Mullen in the IAB Genomic Core Laboratory.

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
