## [Editor Report]

Editor’s AssessmentAntimicrobial resistance (AMR) is a global public health threat, and environmental microbial communities can act as reservoirs for resistance genes. There is a need for genomic surveillance could provide insights into how these reservoirs change and impact public health. With that goal in mind this study tested the ability of nanopore sequencing and adaptive sampling to enrich for AMR genes in a mock community of environmental origin. On average adaptive sampling resulting in a target composition 4x higher than without adaptive sampling, and increased target yield in most replicates. The methods and scripts for this approach were reviewed and curated together, although the scope of this study was limited in terms of communities tested and AMR genes targeted. And the authors improved their analysis by conducting an additional analysis of a diverse microbial community. Demonstrating the method is reusable and its results are promising for developing a flexible, portable, and cost-effective AMR surveillance tool.

---

## [Reviewer Report]

Reviewer name and names of any other individual's who aided in reviewerNed PeelDo you understand and agree to our policy of having open and named reviews, and having your review included with the published manuscript. (If no, please inform the editor that you cannot review this manuscript.)YesIs the language of sufficient quality?YesPlease add additional comments on language quality to clarify if neededIs there a clear statement of need explaining what problems the software is designed to solve and who the target audience is? YesAdditional CommentsIs the source code available, and has an appropriate Open Source Initiative license <a href="https://opensource.org/licenses" target="_blank">(https://opensource.org/licenses)</a> been assigned to the code?YesAdditional CommentsI do not think the authors have included a specific license and assume the code will be released under a Creative Commons CC0 waiver.As Open Source Software are there guidelines on how to contribute, report issues or seek support on the code?NoAdditional CommentsNo guidelines on how to contribute, report issues or seek support on the code.Is the code executable?YesAdditional CommentsIs installation/deployment sufficiently outlined in the paper and documentation, and does it proceed as outlined?YesAdditional CommentsIs the documentation provided clear and user friendly?YesAdditional CommentsIs there enough clear information in the documentation to install, run and test this tool, including information on where to seek help if required?YesAdditional CommentsIs there a clearly-stated list of dependencies, and is the core functionality of the software documented to a satisfactory level?YesAdditional CommentsA list of software used, along with version numbers, can be found in "dart_methods_notebook.md"Have any claims of performance been sufficiently tested and compared to other commonly-used packages? YesAdditional CommentsIs test data available, either included with the submission or openly available via cited third party sources (e.g. accession numbers, data DOIs)?YesAdditional CommentsAre there (ideally real world) examples demonstrating use of the software? YesAdditional CommentsIs automated testing used or are there manual steps described so that the functionality of the software can be verified?YesAdditional CommentsThe authors describe each step of the analysis well and have provided code to reproduce the analysis and figures from the manuscript.Any Additional Overall Comments to the AuthorRecommendationAccept

---

## [Reviewer Report]

Reviewer name and names of any other individual's who aided in reviewerJulian SommerDo you understand and agree to our policy of having open and named reviews, and having your review included with the published manuscript. (If no, please inform the editor that you cannot review this manuscript.)YesIs the language of sufficient quality?YesPlease add additional comments on language quality to clarify if neededIs there a clear statement of need explaining what problems the software is designed to solve and who the target audience is? NoAdditional CommentsNot applicable to this study, since no novel software is described.Is the source code available, and has an appropriate Open Source Initiative license <a href="https://opensource.org/licenses" target="_blank">(https://opensource.org/licenses)</a> been assigned to the code?NoAdditional CommentsNot applicable to this study, since no novel software is described.As Open Source Software are there guidelines on how to contribute, report issues or seek support on the code?NoAdditional CommentsNot applicable to this study, since no novel software is described.Is the code executable?Unable to testAdditional CommentsThe code and software used for analysis of the data is reported in the supplement data. However, the data used in this study in the SRA biobank is not available to download at the time of this review.Is installation/deployment sufficiently outlined in the paper and documentation, and does it proceed as outlined?Unable to testAdditional CommentsSee aboveIs the documentation provided clear and user friendly?YesAdditional CommentsThe analysis steps are clearly commented.Is there enough clear information in the documentation to install, run and test this tool, including information on where to seek help if required?NoAdditional CommentsThe code provided for the data analysis is not usable without the raw sequencing data.Is there a clearly-stated list of dependencies, and is the core functionality of the software documented to a satisfactory level?YesAdditional CommentsHave any claims of performance been sufficiently tested and compared to other commonly-used packages? Not applicableAdditional CommentsIs test data available, either included with the submission or openly available via cited third party sources (e.g. accession numbers, data DOIs)?NoAdditional CommentsAre there (ideally real world) examples demonstrating use of the software? NoAdditional CommentsIs automated testing used or are there manual steps described so that the functionality of the software can be verified?NoAdditional CommentsAny Additional Overall Comments to the AuthorThe aim of this study was to test the ability of adapting sampling sequencing on the Oxford Nanopore sequencer to enrich for antibiotic resistance genes in a synthetic mixture of bacterial DNA. DNA from six environmental bacterial isolates with known antibiotic resistance genes were mixed at equal mass and used for metagenomic sequencing on an Oxford Nanopore MinION MK1B, comparing adaptive sampling with standard sequencing. By analysing 10 sequencing runs using low throughput, low cost flongle flow cells, the authors obtained sequencing data to compare adaptive sampling and standard sequencing approaches. Using a defined composition of sequenced sample and technical and biological replicates, the method is generally suitable. From their data, the authors conclude that adaptive sequencing significantly reduces throughput and increases gene target enrichment by analysing different parameters.  This result is important for the use of adaptive sampling in general, but has already been published in numerous publications, the author cites in his study. According to the author, the novel aspect of this work is the environmental origin of the bacteria used to generate the synthetic mock community. However, since the approach of adaptive sampling does not change regardless of the origin of the sequenced DNA, there are no significant new insights generated in this study. Also, the synthetic mock community of six members does not resemble an environmental metagenomic sample with incomparably more complex species diversity with different abundances. From the data presented in this study, no conclusions can be drawn regarding the performance of adaptive sampling sequencing of environmental metagenomic samples.  To improve the study, I suggest the following: Sequencing of DNA from environmental samples using nanopore sequencing without adaptive sampling and identification of antibiotic resistance genes. Subsequently, resequencing the sample using adaptive sampling based on the identified antibiotic resistance genes and comparing the results in terms of gene target enrichment as analysed in the study. This was partly suggested by the authors and should be carried out to gain new insights into the very interesting application of metagenomic sequencing for the One Health approach. Additionally, there are some inconsistencies in the manuscript. For example, line 128 – 132 describes the sequencing process using different flowcells and technical replicates. However, it remains unclear, how the half of the channels of each flowcell were reserved for adaptive sampling sequencing since the adaptive sampling sequencing is always performed on the whole flowcell. Additionally, it is stated, that each flowcell was used twice for sequencing, however, no method on how to reuse the flongle flowcells is described and no protocol for this is available from oxford nanopore.
RecommendationReject (Unsound or Unusable)